# Classification System of the Sagittal Integral Morphotype in Children from the ISQUIOS Programme (Spain)

**DOI:** 10.3390/ijerph17072467

**Published:** 2020-04-04

**Authors:** Fernando Santonja-Medina, Mónica Collazo-Diéguez, María Teresa Martínez-Romero, Olga Rodríguez-Ferrán, Alba Aparicio-Sarmiento, Antonio Cejudo, Pilar Andújar, Pilar Sainz de Baranda

**Affiliations:** 1Department of Medicine and Orthopaedic Surgery, Faculty of Medicine, Regional Campus of International Excellence “Campus Mare Nostrum”, University of Murcia, C.P. 30100 Murcia, Spain; santonja@um.es; 2Sports and Musculoskeletal System Research Group (RAQUIS), University of Murcia, C.P. 30100 Murcia, Spain; monicacodi@hotmail.com (M.C.-D.); olga.rodriguez@um.es (O.R.-F.); alba.aparicio@um.es (A.A.-S.); antonio.cejudo@um.es (A.C.); psainzdebaranda@um.es (P.S.d.B.); 3Department of Rehabilitation Sciences and Physiotherapy, Albacete University Hospital Complex, C.P. 02006 Albacete, Spain; pilarandujar.albacete@gmail.com; 4Department of Physical Activity and Sport, Faculty of Sport Sciences, Regional Campus of International Excellence “Campus Mare Nostrum”, University of Murcia, C.P. 30720 Murcia, Spain

**Keywords:** assessment, spine, sagittal morphotype, school-aged

## Abstract

The sagittal spinal morphology presents 4 physiological curvatures that increase endurance to axial compression forces and allow adequate postural balance. These curves must remain within normal ranges to achieve a static and dynamic balance, a correct functioning of the muscles and an adequate distribution of the loads, and thus minimize the injury risk. The purpose of this study was to categorize the sagittal spinal alignment according to the different morphotypes obtained for each curve in standing, slump sitting, and trunk forward bending positions in schoolchildren. It was a cross-sectional study. Sagittal spinal curvatures were assessed in 731 students from 16 elementary schools. In the sagittal standing position assessment, 70.45% and 89.06% of schoolchildren presented a “normal” morphotype for both dorsal and lumbar curves, respectively. After the application of the “Sagittal Integral Morphotype” protocol according to the morphotypes obtained in the three positions assessment (standing, slump sitting, and trunk forward bending), it was observed how the frequency of normal morphotypes for the dorsal and lumbar curve decreased considerably (only 32% and 6.6% of children obtained a “normal sagittal integral morphotype” for the thoracic and lumbar curvatures, respectively). These results show how it is necessary to include the slump sitting and trunk forward bending assessment as part of the protocol to define the “integral” sagittal alignment of the spine and establish a correct diagnosis. The use of the diagnostic classification presented in this study will allow early detection of misalignment not identified with the assessment of standing position.

## 1. Introduction

The sagittal spinal morphology presents 4 physiological curvatures (cervical and lumbar lordosis, thoracic and sacral kyphosis) that increase endurance to axial compression forces and allow adequate postural balance [1]. These curves must remain within normal ranges to achieve a static and dynamic balance, a correct functioning of the muscles, and adequate distribution of the loads, and thus minimize the injury risk [2]. 

Several studies have quantified the sagittal spinal alignment in adults and adolescents [3,4,5,6,7,8,9,10]. However, few studies have performed a classification according to sagittal spinal alignment in children and adolescents [11,12,13,14].

Smith et al. [13] quantified three angular measures of the sagittal plane in standing position calculated from lateral photographs with retro-reflective markers placed on bony landmarks and defined 4 morphotypes: (1) “sway-back” or hyperkyphotic morphotype, (2) “flat-back” or flat morphotype, (3) “neutral” or normal morphotype, and (4) “hyperlordotic” or hyperlordotic morphotype.

Dolphens et al. [11,12] also used photographs during habitual standing to develop a classification system according to 3 gross postural (pelvic displacement, trunk lean, and body lean angle) and 5 lumbopelvic characteristics (pelvic tilt, sacral inclination, lumbar lordosis, lumbar apex, and number of lordotic vertebrae). Cluster analysis indicated 3 types of characteristic overall sagittal profiles: (1) “neutral global alignment” or normal morphotype, (2) “sway-back” or hyperkyphotic morphotype, and (3) “leaning-forward” morphotype.

The assessment of the morphotype in standing position has been the most used to classify the sagittal spinal alignment, however, this will reveal a single position of the many that can be adopted in daily life. Therefore, Stagnara [15], Bradford [16], Chopin and David [17] highlighted the importance of assessing the trunk bending forward and sitting positions within the assessment of the spine as a complement to the sagittal plane study. 

Bado [18] was the first author to highlight the importance of trunk bending forward and describe a new misalignment that he called “functional kyphosis”. This “dynamic” misalignment is observed when there is a normal alignment of the spine in the standing position but appears an increase of the dorsal curvature in the trunk bending forward position.

On the other hand, Santonja and Pastor [19] described the “lumbar kyphotic attitude” for the first time, which consists of excessive lumbar kyphosis when trunk bending forward and/or when sitting, but with normal alignment of the lumbar spine in the standing position. While Somhegyi and Ratko [20] proposed the term “lumbar hypermobility” for those cases where there is a lumbar hyperlordosis during standing together with excessive lumbar kyphosis in the trunk bending forward and/or sitting positions.

The “lumbar kyphotic attitude” is a morphotype frequently adopted by schoolchildren and today its future repercussion is not known, nor its relation to low back pain and disc degeneration. The study of the sagittal spinal alignment during sitting is important because it is one of the positions that people adopt for many daily activities and, therefore, can have a significant influence on the development of spinal morphology [21].

The detection of these altered dynamic morphotypes acquires importance, especially during growth, before the peak height velocity [11,12,22,23]. For example, if it is not diagnosed in time, it could be produced an increase of the thoracic curvature towards a hyperkyphotic spine with a tendency to structuring [23].

For that reason, Santonja [24] proposed the assessment of the “Sagittal Integral Morphotype” which included the three positions: standing, slump sitting, and trunk bending forward. Therefore, the combination of the different spinal morphotypes in each of these positions will allow an accurate diagnosis of the sagittal misalignments of the spine. 

The assessment of the spinal curvatures in the standing, sitting, and flexion positions of the trunk has been the subject of several studies [25,26,27,28,29], due to the effects that these postures can have on the development of the sagittal plane of the spine [30,31]. A greater thoracic or lumbar kyphosis has been associated with an increase in vertebral stress [32] and greater intradiscal pressure in the thoracic and lumbar intervertebral discs [33,34,35,36], which increase the risk of spinal injury, while a greater lumbar lordosis can result in large changes in the distribution of loading on the posterior elements of the vertebra (e.g., facet joints, neural arch) [37]. 

Proper posture is increasingly less present in the everyday life of children. For it, systematic monitoring and assessment of the postural status of children is essential and can detect many health problems in time before they become serious [38]. Therefore, the main objective of the present study was to characterize the posture in the sagittal plane and to define the “Sagittal Integral Morphotype” taking into account the classification of the sagittal spinal alignment according to the different morphotypes obtained for each curve in standing, slump sitting, and trunk forward bending positions in schoolchildren aged 7 to 13 years.

## 2. Materials and Methods 

### 2.1. Design

The presented study was a descriptive study. Before participation in a postural hygiene program, angular values for sagittal spinal curvatures (thoracic and lumbar) of primary school students were recorded in a relaxed standing, slump sitting position, and trunk forward bending positions in order to describe the “Sagittal Integral Morphotype” in schoolchildren. 

### 2.2. Participants

First, a total of 887 students were selected through a convenience sample from several elementary schools that had been selected to participate in the ISQUIOS Program, a postural hygiene program which is carried out in the region of Murcia, Spain. 

As inclusion criteria, those who were in 3rd–6th grade and were from 8 to 12 years old (a), who attended the day of the assessment (b), and who delivered the signed written consent (c) were included (*n* = 812). However, those who refused to take part on the examination day (a), who had suffered an important physical injury which limited the correct performance of the test (b), who had been previously diagnosed to have scoliosis (c), or who had previously received treatment for any frontal or sagittal plane-related pathology through the use of a corset or specific kinesiotherapy (d), or who showed tactile hypersensitivity resulting in an inability of adopting free, habitual posture (e) were excluded (*n* = 71). Besides, to avoid that outliers could mask the real results, those specific cases were excluded from the final sample (*n* = 10).

Finally, 731 students (females: 379; males: 352) from 16 schools participated in the study (age: 10.55 ± 1.11 years, height: 142.49 ± 8.66 cm, weight: 40.23 ± 10.57 kg). Following the Declaration of Helsinki, the protocol of this study was approved by the Ethics and Research Committee of the University of Murcia (Spain; Protocol Number 77/2013). Therefore, all the students and legal tutors were informed of the procedure and objectives of the study and expressed written consent.

### 2.3. Procedures

Students were instructed not to participate in any training or physical activity 24 h before their assessment [25]. All the measurements were performed on the same day, starting with anthropometric measurements. Body height was measured with the Seca 213 mobile stadiometer (SECA 213, Hamburg, Germany), with an accuracy of 0.1 cm. Body mass was measured using the electronic scale OMRON BF 500 (Omron Healthcare, Inc USA), with an accuracy of 0.1 kg. The measurements were performed in standard conditions. Each subject was evaluated by the same examiner in a single session and participants were asked to take off their shoes and only wear their undergarments during the assessment. Students did not perform warm-up or stretching exercises before or during the measurement [28,39]. The measurements were performed in random order. There was a 5 min rest between the different tests [39]. Three trials for each measure were administered/recommended. When two of those measures were equal, that value was chosen. When the three measures were different, the average value of the two similar measurements was taken for data analysis. Furthermore, it is very important to note that a test trial was carried out before the first measurement with the objective that students were well informed and were sure about how to perform the test. The study was rigorously controlled by keeping the expert and the students blinded to the objective of the study. 

Body posture was examined using inclinometer techniques (ISOMED Unilevel inclinometer, Portland, OR, United States) which are reliable [40], handy, and affordable; therefore, it is frequently used in examinations [41,42,43]. The research methodology was performed following the guidelines in topographic points [44,45]. Following the palpation of the points (spinous processes, transition of kyphosis into lordosis, and posterior iliac spines), we marked it with a dermatograph [31,41,46]. Specifically, before data collection, the spinous process of the first thoracic vertebra (T1), twelfth thoracic vertebra (T12), and fifth lumbar vertebra (L5-S1) were marked on the children’s skin [47,48,49,50,51]. 

The most reliable technique to quantify kyphosis and lordosis is the conventional spinal X-ray method. However, the limitations of radiographic measurement, including cost [52], limited portability, time-consuming, and exposure to ionizing radiation [53,54], make it unsuitable for use in clinical practice.

There are other methods free of ionizing radiation that assess the curvatures of the spine in the sagittal plane, for instance, the inclinometer provides a noninvasive evaluation with good reproducibility, reliability, and correlation with the radiographic measurement [51,55,56,57].

To perform precise measurements of the protocol, we instructed the students to stand comfortably in normal relaxed standing and sitting position and to look straight ahead. Marks on the floor and in front of a wall ensured that all subjects were in the same place and the same position. To measure the slump sitting position, the participant was sitting on a stretcher in a relaxed posture with the forearms resting on the thighs, knees flexed, and without feet support. To measure the trunk forward bending position, the students were required to sit with their knees straight, the legs together and the soles of the feet positioned flat against the end of a sit-and-reach box (height: 32 cm) [55,56,57,58,59].

Furthermore, a double-blind study was conducted before the measurements to establish the tester’s reliability with 12 participants, and intraclass correlation coefficients (ICC) greater than 0.95 were obtained for all variables.

#### 2.3.1. Sagittal Integral Morphotype Assessment

The measuring protocol of the “Sagittal Integral Morphotype”, described by Santonja [24] for the complete evaluation of sagittal spinal curvatures (dorsal and lumbar), consists of the sagittal assessment in a relaxed standing position (SP), in a slump sitting position (SSP), as well as in trunk forward bending position (TFB). The idea of this protocol is to assess the main positions that you can use and adopt in daily and sports activities. Essentially, posture characteristics that can have clinical relevance are quantified using a screening protocol with clinical applicability and are incorporated into a consistent system in which the clinical relevance of the identified posture types is appreciated in terms of their association with the risk of spinal pain and spinal injury. This protocol has been previously used in other studies [49,51,56,57]. First, it is necessary to assess the children in the three positions, and then use the three results to determine and define the “Sagittal Integral Morphotype” in each curve. Negative values stand for degrees of posterior concavity (lordosis), and positive values stand for anterior concavity or kyphosis.

##### Standing Position Assessment (SP)

To assess the SP, the participant was standing and relaxed (with the eyes and ears in line horizontally, arms hanging down laterally to the body, extended knees, and feet shoulder-width apart) [49,51]. The inclinometer was placed at the first mark (T1) and calibrated to 0°, and then the curvature was outlined until maximum angulation of thoracic curvature was reached and the angle was recorded. Subsequently, at this point, the inclinometer was calibrated to 0° again, and the lumbar curvature was outlined until the maximum lumbar angle was reached and recorded. The legs on the inclinometer were adjusted to cradle the spinous processes and were pressed gently but firmly into the interspinal spaces [25]. Negative values stand for degrees of posterior concavity (lordosis), and positive values stand for anterior concavity or kyphosis.

##### Slump Sitting Position Assessment (SSP)

To measure the SSP, the participant was sitting on the stretcher in a relaxed posture with the hands resting on the thighs and without feet support [29,49,51]. For the thoracic curve assessment, the inclinometer was placed at the first mark (T1) and it was calibrated to 0°. Then, the inclinometer was placed on the second mark (T12) and the degrees were recorded. For the lumbar curve, the inclinometer was calibrated to 0° again on this last point and then the inclinometer was placed on the third mark (L5-S1) and the degrees were registered. The legs on the inclinometer were adjusted to cradle the spinous processes and were pressed gently but firmly into the interspinal spaces [25]. Negative values corresponded to lumbar lordosis (posterior concavity) and positive values corresponded to lumbar kyphosis.

##### Trunk Forward Bending Assessment (TFB) during “Sit and Reach Test”

Participants were asked to perform the sit and reach test and keeping the maximum trunk forward bending for 6–8 s while sagittal spinal curvatures were measured following the same procedure as in the SSP [24,46,56,60]. For the thoracic curve assessment, the inclinometer was placed at the first mark (T1) and it was calibrated to 0°. Then, the inclinometer was placed on the second mark (T12) and the degrees were recorded. For the lumbar curve, the inclinometer was calibrated to 0° again on this last point and then the inclinometer was placed on the third mark (L5-S1) and the degrees were registered. The legs on the inclinometer were adjusted to cradle the spinous processes and were pressed gently but firmly into the interspinal spaces [25]. Negative values stand for degrees of posterior concavity (lordosis), and positive values stand for anterior concavity or kyphosis.

#### 2.3.2. References of Normality for Thoracic and Lumbar Curves

The values used in previous studies [49,51,56,57] were used to classify the results related to the thoracic and lumbar curvature in each assessed position. The references of normality are described in Table 1. The concepts that underlie this grading system were derived from clinical and scientific publications [33,34,35,36] as well as from clinical experience.

#### 2.3.3. Diagnostic Classification of the “Sagittal Integral Morphotype”

Table 2 and Table 3 show the different classifications and subclassifications for the integral diagnosis of the sagittal thoracic and lumbar morphotype, respectively, based on the assessment and classification of both curvatures in each position (SP, SSP, and TFB). Thus, the final diagnosis is defined by the morphotype obtained in each position. All subclasses are concerning the position or positions where misalignment occurs. Specifically, the “static” subclasses are about the slump sitting position or to the standing position or in both positions; the “dynamic” subclasses are related to the forward bending position, while the “total” subclasses are referred to when the misalignment occurs in the slump sitting position and in the forward bending position. Normality and pathology values were defined by Santonja [61] after conducting a clinical-radiological study of the sagittal spine and have been used in prior studies [49,51,56,57]. These values are of great importance, as they allow performing a more accurate diagnosis through the use of non-invasive methods.

### 2.4. Statistical Analysis

Before the statistical analysis, the distribution of raw data sets was checked using the Kolmogorov–Smirnov test to determine normal distribution. Descriptive statistics including mean values and standard deviation (SD) as well as absolute and relative frequency were calculated for each classification and subclassification according to their “Sagittal Integral Morphotype”. Besides, the Pearson chi-squared test was used to determine the differences in the frequencies of normality and sagittal imbalance by sex. The analysis was performed using SPSS version 20.0 (SPSS Inc., Chicago, IL, USA). The level of significance was set at *p* < 0.05.

## 3. Results

Table 4 shows the absolute and relative frequency of children in each classification by assessment position according to the thoracic and lumbar curvature.

In relation to the thoracic curvature, 70.45%, 44.32%, and 85.77% of children presented normal kyphosis in SP, SSP, and TFB, respectively. As for the lumbar curve, 89.06%, 12.72%, and 38.43% of schoolchildren showed values normal values during SP, SSP, and TFB, respectively. In contrast, 87.27% and 61.83% of participants presented increased lumbar curvature in SSP and TFB (hyperkyphosis), respectively.

Table 5 and Table 6 show the absolute and relative frequencies of children for each curvature and classification according to the “Sagittal Integral Morphotype” of the spine [24].

Concerning sagittal integral thoracic morphotype classification (Table 5), 234 children presented a “normal” morphotype (32%), since their values were normal in all 3 measurement positions. Two hundred sixty-nine participants were classified as “functional thoracic hyperkyphosis” (36.8%) because they adopted a normal kyphosis in SP, but an increased kyphosis in SSP (static, 30.2%) or in TFB (dynamic, 2.6%). Two hundred scholars were classified as “hyperkyphosis” (27.4%) since they presented hyperkyphotic curvature in SP and SSP (static, 16.3%) or in TFB (dynamic, 0.8%). There were 32 children who presented “total hyperkyphosis” morphotype, as they are classified as “hyperkyphotic” in the three positions analyzed. Thirteen subjects were defined as “hypomobile kyphosis” morphotype (normal kyphosis in SP and SSP, but hypokyphosis in TFB), and fifteen children as “hypokyphosis or hypokyphotic attitude” (normal kyphosis in SSP and TFB, while hypokyphosis in relaxed SP). There were no significant differences between gender (*X^2^*_(*n* = 731)_ = 8314, *p* = 0.503).

According to sagittal integral lumbar morphotype classification (Table 6), only 6.6% of children showed “normal” morphotype (normal lumbar curvature in all three assessed positions), while 82.3% children were classified as “functional lumbar hyperkyphosis” because they presented normal lordosis in SP, but increased kyphosis in SSP (static, 15%) or in TFB (dynamic, 4.1%), or in both positions (total, 63.2%). Fifty-four schoolchildren were classified as “lumbar hypermobility” (7.4%), presenting hyperlordosis curvature in SP and hyperkyphosis in SSP (static, 6.2%) or in TFB (dynamic, 1.1%). Fourteen participants (1.9) were classified as “hypolordosis” (hypolordosis in SP, but normal curve or hyperkyphosis lumbar in SSP and TFB). Twelve children were diagnosed as “hyperlordotic attitude” because they presented hyperlordosis in SP and normal curve in SSP and TFB. Finally, only one child was classified as “structured hyperlordosis” (lumbar curve increased in all three positions). No participants presented the morphotype “structured lumbar kyphosis” or “lumbar spine with reduced mobility”.

Statistically significant differences were found between genders when sagittal integral lumbar morphotype was analyzed (*X^2^*_(*n* = 731)_ = 42,636, *p* < 0.001). In particular, significantly more males presented “functional lumbar hyperkyphosis” (86.9% vs. 78.1%), while considerably more females showed “lumbar hypermobility” (10.3% vs. 4.3%) or “hyperlordotic attitude” (2.4% vs. 0.9%).

## 4. Discussion

The sagittal spinal misalignments may appear due to an increase (hyperkyphosis, hyperlordosis, and kyphotic-lordotic posture), decrease (flat-back, hypolordosis, and hypokyphosis) or inversion (thoracic lordosis and lumbar kyphosis) of one or both curves or when the normal spine topography is modified (thoracic-lumbar kyphosis, cervical-thoracic kyphosis) [58,59,62,63]. In the current study, 3 morphotypes categories (normal, hypo-, and hyper-) are determined for each curvature and position assessed. This classification system of 3 morphotypes was further subcategorized according to the integral diagnosis based on the combination of the different positions assessed, thereby establishing 12 sagittal subcategories for the dorsal curvature and 16 subcategories for the lumbar curve. Because this is the first study in which these 3 different positions are combined for the diagnosis of the sagittal spine, comparison with existing literature is difficult. Other studies have proposed different classification systems to define the overall sagittal thoraco-lumbo-pelvic alignment but they only assess the morphotype in standing position [11,12,13,14].

In this sense, Smith et al. (2008) [13] found “neutral spine” in 30.2% of participants between 13 to 15 years, defining “neutral posture” as lack of upper trunk displacement and normal lumbar lordosis and thoracic kyphosis. Later, Dolphens et al. (2013, 2014) [11,12] measured children in the pre-peak height velocity (boys at 12 years and girls at 10 years) and found that 40.88% of them had a “neutral” global alignment which was characterized by small pelvic displacement angle, small trunk lean angle, and intermediate body lean angle close to 0°. 

It must be taken into account that these studies only assessed the spine of the students in standing position, but these can adopt other more predominant positions in daily life. Sitting and trunk forward bending positions can have a significant influence on the development of spinal morphology [21]. If only the values obtained in this study for standing position are kept in mind, 70.45% and 89.6% of schoolchildren would have a normal dorsal and lumbar morphotype, respectively, making an erroneous diagnosis of the children’s spine. However, according to the “Sagittal Integral Morphotype” (standing, slump sitting, and trunk forward bending positions) the present results showed that only 32% of schoolchildren presented a normal thoracic kyphosis morphotype, whereas in the normal lumbar lordosis morphotype it decreased to 6.6%.

Regarding the thoracic curve, the most frequent diagnosis is “functional thoracic hyperkyphosis” (36.8%). Logical result considering that schoolchildren are sitting many hours a day (school, homework, video games, TV…), performing less physical activities and also, sitting very often with poor postural hygiene, which leads to hyperkyphotic dorsal spines.

Concerning the lumbar curve, the “functional lumbar hyperkyphosis” was by far the most prevalent diagnosis (82.4%), which was originally described as “lumbar kyphotic attitude” [61], followed by the “lumbar hypermobility” (7.4%), “hypolordosis” (1.9%), and “hyperlordotic attitude” (1.6%). 

When the results of lumbar morphotype were analyzed by gender, it was observed that boys had higher “functional lumbar hyperkyphosis” morphotype (86.9% vs. 78.1%). In contrast, girls showed higher “lumbar hypermobility” (10.3% vs. 4.3%) and “hyperlordotic attitude” (2.4% vs. 0.9%) morphotypes. This gender-specific morphotype was observed in other studies [64] in which girls showed a “hyperlordotic” posture due to increased lumbar angle and boys presented a “sway” posture probably determined by a predominant backward tilt of the spine. The increased lumbar angle among girls could be explained by the structural phylogenetic adaptations developed by the female spine [22,64,65,66]. Therefore, it can be said that different gender patterns exist among schoolchildren due to the gender-specific biomechanical frameworks of the spino-pelvic, implying different biomechanical loads influence the specific development of pediatric spinal deformities by gender, for example, higher prevalence of scoliosis in girls and Scheuermann’s disease in boys [64].

The clinical importance of postural morphotypes is evident [11,13,63,64,67,68,69]. For example, back pain in adults has been related to “flat-back” and lordotic morphotypes [67,68]. During the adolescence period, different authors discovered that those who presented a non-neutral spinal alignment proved to be more likely to suffer from back pain [13], and in children assessed before the peak height velocity, a “sway-back” morphotype was associated with a higher prevalence of neck and low back pain [11]. 

Quantitative evaluation of spinal curvature is valuable for planning of orthopedic surgical procedures, monitoring the progression and treatment of spinal deformities, and for determining reference values in normal and pathological conditions [70]. Spinal curvature is one of the most significant spinal parameters [71], however, the only use of the standing position to assess the sagittal alignment will make that numerous altered morphotypes go unnoticed and, therefore, it can lead to an erroneous diagnosis. An incorrect interpretation of the sagittal spinal alignment can have significant consequences for the patient, not only in terms of deformity and pain but also in preventive and treatment terms [59].

Regular screening of the spine in children is recommended today. The American Academy of Orthopedic Surgeons proposes regular screening of children aged 11 while the American Academy of Pediatricians has proposed regular screening of the backbone of school-age children from 10 years of age [23]. However, Rusnák et al. [72] suggest regular backbone and posture screening in children at the beginning of compulsory school attendance. In this period of the child’s life, the musculoskeletal system is the most overloaded by carrying a school bag, long-term sitting, and a general change of lifestyle. It increases in children in the second year of elementary school (age of 7–8). Children at this age represent a critical group for the development of spinal deformities and postural disorders [23].

Certain limitations of the study should be considered. For the generalization of data obtained in the general population, it is necessary to carry out extensive research as soon as possible, which will include the examination of age-related bad posture [23]. For it, subsequent studies should include and examine the influence of different morphotypes on back pain and musculoskeletal disorders. Future follow-up of back pain in these children will be necessary to improve the knowledge about the clinical role of sagittal spinal alignment and musculoskeletal disorders during growth. Another point not taken into consideration is the maturational difference between boys and girls of similar chronological age. Furthermore, given the variations regarding terminology and criteria across studies, the compelling need for a consensus statement regarding nomenclature should be highlighted. 

The sample size, using validated assessment protocol and data from elementary schoolchildren are the main strengths of this study. This methodology is available, a simple, unexpansive, and reliable diagnostic method. Furthermore, the inclinometer methodology assessment is recommended as a valid instrument for measuring sagittal spinal alignment, with a strong agreement with the gold standard (spinal X-ray method) [73]. Any orthopedist or physician doctor, physiotherapist or sports science expert may use it in their practice to monitor the incidence of spinal deformities and postural disorders and propose timely preventive and therapeutic treatment. We consider this as an advantage of the diagnostic’s methodology used in the present investigation. 

However, since posture is the interaction of muscles, tendons, and bones, muscle activity changes the curvature of the back when standing, bending forward, or sitting. Therefore, the angles of kyphosis and measurable lordosis might consequently change. Thus, for those who want to assess the spinal curvatures in the sagittal plane, it is mandatory to verify the reliability of the protocol and the measurements for each position or movement.

## 5. Conclusions

The use of the diagnostic classification presented in this study will allow early detection of misalignment not identified with the assessment of standing position. These results show how it is necessary to include the slump sitting and trunk forward bending assessment as part of the protocol to define the “integral” sagittal alignment of the spine and establish a correct diagnosis.

In addition, the following statements outline the main findings:In the sagittal standing position assessment, 70.45% and 89.06% of schoolchildren (boys and girls) presented a “normal” morphotype for both dorsal and lumbar curves, respectively.After the application of the “Sagittal Integral Morphotype” protocol according to the morphotypes obtained in the three positions assessment (standing, slump sitting, and trunk forward bending), it was observed how the frequency of normal morphotypes for the dorsal and lumbar curve decreased considerably.It can be observed that only 32% and 6.6% of children obtained a “normal sagittal integral morphotype” for the thoracic and lumbar curvatures, respectively.For the thoracic spine, the most common morphotype was “functional thoracic hyperkyphosis” (36.8%), without differences by sex.For the lumbar spine, the most common morphotype was “functional lumbar hyperkyphosis” (82.3%). Sex differences were found, concretely, males presented higher cases of “functional lumbar hyperkyphosis” (86.9% vs. 78.1%), and females showed higher percentages of “lumbar hypermobility” (10.3% vs. 4.3%) and “hyperlordotic attitude” (2.4% vs. 0.9%).

## Figures and Tables

**Table 1 ijerph-17-02467-t001:** Values of normality for thoracic and lumbar curves in each position [49,51].

Spinal Curve	SP ^1^		SSP ^2^		TFB ^3^	
	Classification	Values	Classification	Values	Classification	Values
Thoracic	Hypokyphosis	<20°	Hypokyphosis	<20°	Hypokyphosis	<40°
	Normal	20° to 40°	Normal	20° to 40°	Normal	40° to 65°
	Hyperkyphosis	>40°	Hyperkyphosis	>40°	Hyperkyphosis	>65°
Lumbar	Hypolordosis	<−20°	Hyperlordosis	<-15°	Hypokyphosis	<10°
	Normal	−20° to −40°	Normal	−15 to 15°	Normal	10° to 30°
	Hyperlordosis	>−40°	Hyperkyphosis	>15°	Hyperkyphosis	>30°

^1^ SP: Standing position; ^2^ SSP: Slump sitting position; ^3^ TFB: Trunk forward bending.

**Table 2 ijerph-17-02467-t002:** Diagnostic classification of the “Sagittal Integral Morphotype” for the thoracic curve.

Classification	Subclassification	SP ^1^	SSP ^2^	TFB ^3^
Normal kyphosis		Normal (20°–40°)	Normal (20°–40°)	Normal (40°–65°)
Functional Thoracic Hyperkyphosis	Static	Normal (20°–40°)	Hyperkyphosis (>40°)	Normal (40°–65°)
	Dynamic	Normal (20°–40°)	Normal (20°–40°)	Hyperkyphosis (>65°)
	Total	Normal (20°–40°)	Hyperkyphosis (>40°)	Hyperkyphosis (>65°)
Hyperkyphosis	Total	Hyperkyphosis (>40°)	Hyperkyphosis (>40°)	Hyperkyphosis (>65°)
	Standing	Hyperkyphosis (>40°)	Normal (20°–40°)	Normal (40°–65°)
	Static	Hyperkyphosis (>40°)	Hyperkyphosis (>40°)	Normal (40°–65°)
	Dynamic	Hyperkyphosis (>40°)	Normal (20°–40°)	Hyperkyphosis (>65°)
Hypokyphosis/ Hypokyphotic attitude	Flat-back	Hypokyphosis (<20°)	Hypokyphosis (<20°)	Hypokyphosis (<40°)
	Standing	Hypokyphosis (<20°)	Normal (20°–40°)	Normal (40°–65°)
	Static	Hypokyphosis (<20°)	Hypokyphosis (<20°)	Normal (40°–65°)
	Dynamic	Hypokyphosis (<20°)	Normal (20°–40°)	Hypokyphosis (<40°)
Hypomobile kyphosis		Normal (20°–40°)	Normal (20°–40°)	Hypokyphosis (<40°)

^1^ SP: Standing position; ^2^ SSP: Slump sitting position; ^3^ TFB: Trunk forward bending.

**Table 3 ijerph-17-02467-t003:** Diagnostic classification for the “Sagittal Integral Morphotype” for the lumbar curve.

Classification	Subclassification	SP ^1^	SSP ^2^	TFB ^3^
Normal lordosis		Normal (−20°/−40°)	Normal (0°±15°)	Normal (10°–30°)
Lumbar spine with reduced mobility	Functional lumbar lordosis // Hypomobile lordosis	Normal (−20°/−40°)	Normal (0°±15°)	Hypokyphosis or lordosis (<10°)
	Lumbar hypomobility	Hypolordosis (<−20°)	Normal (0°±15°)	Hypokyphosis (<10°)
Hyperlordotic attitude		Hyperlordosis (>−40°)	Normal (0°±15°)	Normal (10°–30°)
Functional lumbar hyperkyphosis	Static	Normal (−20°/−40°)	Hyperkyphosis (>15°)	Normal (10°–30°)
	Dynamic	Normal (−20°/−40°)	Normal (0°±15°)	Hyperkyphosis (>30°)
	Total	Normal (−20°/−40°)	Hyperkyphosis (>15°)	Hyperkyphosis (>30°)
Lumbar Hypermobility	Hypermobility 1	Hyperlordosis (>−40°)	Hyperkyphosis (>15°)	Hyperkyphosis (>30°)
	Hypermobility 2	Hyperlordosis (>−40°)	Normal (0±15°)	Hyperkyphosis (>30°)
	Hypermobility 3	Hyperlordosis (>−40°)	Hyperkyphosis (>15°)	Normal (10°–30°)
Hypolordosis	Hypolordotic attitude	Hypolordosis (<−20°)	Normal (0±15°)	Normal (10°–30°)
	Lumbar kyphosis 1	Hypolordosis (<−20°)	Hyperkyphosis (>15°)	Hyperkyphosis (>30°)
	Lumbar kyphosis 2	Hypolordosis (<−20°)	Hyperkyphosis (>15°)	Normal (10°–30°)
	Lumbar kyphosis 3	Hypolordosis (<−20°)	Normal (0°±15°)	Hyperkyphosis (>30°)
Structured Hyperlordosis		Hyperlordosis (>−40°)	Hyperlordosis (<−15°) or normal (0°±15°)	Lordosis or Hypokyphosis (<10°)
Structured lumbar kyphosis		Hypolordosis or kyphosis (<−20°)	Hyperkyphosis (>15°)	Hyperkyphosis (>30°)

^1^ SP: Standing position; ^2^ SSP: Slump sitting position; ^3^ TFB: Trunk forward bending.

**Table 4 ijerph-17-02467-t004:** Mean (SD), absolute and relative frequency of children in each classification by assessment position for each spinal curve according to normality values.

Curvature	Position	Classification	Mean ± SD	n	%
Thoracic curve	SP ^1^	Rectification (<20°)	15.75 ± 2.62°	16	2.18
		Normal (20° to 40°)	32.6 ± 5.9°	515	70.45
		Hyperkyphosis (≥41°)	46.85 ± 4.45°	200	27.36
	SSP ^2^	Hypokyphosis (<20°)	16.67 ± 2.31°	3	0.41
		Normal (20° to 40°)	35.29 ± 5.48°	324	44.32
		Hyperkyphosis (≥41°)	49.15 ± 5.42°	404	55.27
	TFB ^3^	Hypokyphosis (<40°)	32.22 ± 6.71°	18	2.46
		Normal (40° to 65°)	53.88 ± 6.55°	627	85.77
		Hyperkyphosis (≥66°)	70.56 ± 5.93°	86	11.77
Lumbar curve	SP ^1^	Rectification (<-20°)	−16.43 ± −1.78°	14	1.89
		Normal (−20° to −40°)	−31.05 ± −5.97°	650	89.06
		Hyperlordosis (≥−41°)	−45.94 ± −3.09°	67	9.05
	SSP ^2^	Hyperlordosis (<−15°)	-	0	0
		Normal (−15° to 15°)	9.76 ± 3.7°	93	12.72
		Hyperkyphosis (≥16°)	26.91 ± 6.47°	638	87.27
	TFB ^3^	Hypokyphosis (<10°)	9 ± 1.41°	2	0.27
		Normal (10° to 30°)	26.08 ± 4.23°	277	38.43
		Hyperkyphosis (≥31°)	37.96 ± 5.03°	452	61.83

^1^ SP: Standing position; ^2^ SSP: Slump sitting position; ^3^ TFB: Trunk forward bending.

**Table 5 ijerph-17-02467-t005:** Absolute and relative frequency of children in each classification and subclassification of “Sagittal Integral Morphotype” for the thoracic curve.

Classification	Subclassification	Boys (*n* = 352)	Girls (*n* = 379)	Total (*n* = 731)
Normal Kyphosis		112 (31.8%)	122 (32.2%)	234 (32%)
Functional Thoracic Hyperkyphosis	Static	113 (32.1%)	108 (28.5%)	221 (30.2%)
	Dynamic	6 (1.7%)	13 (3.4%)	19 (2.6%)
	Total	12 (3.4%)	17 (4.5%)	29 (4%)
Hyperkyphosis	Total	12 (3.4%)	20 (5.3%)	32 (4.4%)
	Standing	25 (7.1%)	18 (4.7%)	43 (5.9%)
	Static	53 (15.1%)	66 (17.4%)	119 (16.3%)
	Dynamic	3 (0.9%)	3 (0.8%)	6 (0.8%)
Hypokyphosis/Hypokyphotic attitude	Flat-Back	-	-	-
	Standing	8 (2.3%)	7 (1.8%)	15 (2.1%)
	Static	-	-	-
	Dynamic	-	-	-
Hypomobile kyphosis		8 (2.3%)	5 (1.3%)	13 (1.8%)

**Table 6 ijerph-17-02467-t006:** Absolute and relative frequency of children in each classification and subclassification of “Sagittal Integral Morphotype” for the lumbar curve.

Classification	Subclassification	Boys (*n* = 352)	Girls (*n* = 379)	Total (*n* = 731)
Normal lordosis		19 (5.4%)	29 (7.7%)	48 (6.6%)
Lumbar spine with reduced mobility	Functional lumbar lordosis/Hypomobile lordosis	-	-	-
	Lumbar hypomobility	-	-	-
Hyperlordotic attitude		3 (0.9%)	9 (2.4%)	12 (1.6%)
Functional lumbar hyperkyphosis	Static	71 (20.2%)	39 (10.3%)	110 (15%)
	Dynamic	5 (1.4%)	25 (6.6%)	30 (4.1%)
	Total	230 (65.3%)	232 (61.2%)	460 (63.2%)
Lumbar Hypermobility	Hypermobility 1	13 (3.7%)	32 (8.4%)	45 (6.2%)
	Hypermobility 2	-	1 (0.3%)	1 (0.1%)
	Hypermobility 3	2 (0.6%)	6 (1.6%)	8 (1.1%)
Hypolordosis	Hypolordotic attitude	-	-	-
	Lumbar kyphosis 1	7 (2%)	4 (1.1%)	11 (1.5%)
	Lumbar kyphosis 2	2 (0.6%)	-	2 (0.3%)
	Lumbar kyphosis 3	-	1 (0.3%)	1 (0.1%)
Structured Hyperlordosis		-	1 (0.3%)	1 (0.1%)
Structured lumbar kyphosis		-	-	-

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
