# Peer review of "Classification System of the Sagittal Integral Morphotype in Children from the ISQUIOS Programme (Spain)"

_ijerph, 2020, doi:10.3390/ijerph17072467_

Round 1
Reviewer 1 Report
The aim of the present study was to categorize the sagittal spinal alignment according to different morphotypes in different positions (standing, slump sitting and trunk forward bending) in pupils aged 7 to 13 years. Unfortunately, the presentation of the study in the present manuscript has too many shortcomings to justify publication in my opinion.
Introduction:
The introduction to the study first presents some historical approaches to measuring different parameters of spinal alignment. This is comprehensible up to the point where the authors introduce the "Sagittal Integral Morphotype". They refer to a study by Santonja (1996), which is only available in Spanish. At this point, the objective of the study, but also the exact course of the assessment is no longer comprehensible. The basic models for the categorization of spinal morphology are only referenced from studies in Spanish language. Some of them are obviously not internationally published studies, but rather doctoral theses.
Methods:
The theoretical approaches to classify spinal curvatures are therefore not comprehensible to the reader, nor is the exact execution of the assessment.
How the "normative values" on which the authors base their work come into place is also not comprehensible, since the Spanish study of Santonja seems to be the basis again.
In addition, I view the measurement of the spinal curvature with an inclinometer critically, since in children the back contour curvature changes reflectively already by touching the back. For this reason, many other studies rely on non-contact measurement methods.
Discussion:
A total of 13 of the cited literature references are given in Spanish. This does not allow a reproducibility of the theoretical approaches of the study for the reader.
Finally, the study is only descriptive in character, describing the prevalence of certain back contours (lordosis, kyphosis) in 3 different positions. However, since the underlying classification of these forms is unclear (and not comprehensible to the reader in English), no interpretation can be derived. A possible conclusion for medical or health practice therefore remains unclear.
Reviewer 2 Report
Lack of conclusion - in abstract. The sentence at the beginning of the discussion is unnecessary The objective of the present study was to categorize the sagittal spinal alignment according to 222 the different morphotypes obtained for each curve in standing, slump sitting and trunk forward 223 bending positions in schoolchildren aged 7 to 13 years. The authors could include the chapter entitled Limitation of this study - at the end.
There are too few current literature items in the bibliography.
Reviewer 3 Report
Overall this is a well written paper with an interesting result in a large sample. The results are based on rational working hypothesis
INTRODUCTION
The introduction provides sufficient background information for readers to understand the problem. The introduction provides a good perspective of the main topic.
Motivations for this study are more than clear. The objectives are clearly defined at the Introduction, the argumentation was concise and clarifying.
METHODS
They are according for the research aim
RESULTS
All of the tables include specific and good developed statistics.
DISCUSSION
All possible interpretations of the data considered are consistent.
Include a limitation and practical application section
The conclusions have coherence with the initial hypothesis, in addition, they are well established and according to the present discussion.
LITERATURE CITED
The literature cited is relevant to the study, but there are several instances, which have been noted above, in which the author makes assertions without substantiating them with references, but which are sustained by the main text and previous citations.
SIGNIFICANCE AND NOVELTY
As it stands, the results are novel and important enough for this journal.
Round 2
Reviewer 1 Report
Dear authors,
Thank you for the revision of your manuscript, which is now easier to understand. In addition to the points listed below, I would like to ask you to take a critical look at the following aspect in your discussion:
Posture is the interaction of bones, tendons and muscles. When standing and sitting, muscular activity changes the curvature of the back and thus also the measurable kyphosis and lordosis angles. You should explain in more detail how you were able to ensure that the children did not change their trunk posture due to muscular activation during the measurements. I think that this should be clearly addressed in the limitation section, because it is a limiting factor that we generally have in posture measurements (especially in children) and that can limit the accuracy of a classification of the measurement results.
line 91: "...injury. While..." --> "...injury, while..."
line 103: "It was a descriptive study." --> "The presented study was a descriptive study."
line 206: "...Santonja [14]after..." --> "...Santonja [14] after..."
Table 2 & 3: what is the difference between the 'static' and 'dynamic' subclasses? I cannot find an explanation in the text.
line 286: "...these studies only assessment..." --> "...these studies only assessed..."
line 362: "...70.45% and 89.06% of schoolchildren..." - boys and girls?
Author Response
Dear Miki Shen and reviewer 1,
Thank you for the comments and suggestions proposed for the article “Classification System of the Sagittal Integral Morphotype in Children from the ISQUIOS Programme (Spain)”.
We are pleased to provide a cover letter to explain *point-by-point* the
details of the revisions in the manuscript and our responses to the
reviewers' comments. Any revisions are “clearly highlighted” with the “Track Changes” function in Microsoft Word.
Explanation point by point:
Point 1: Thank you for the revision of your manuscript, which is now easier to understand.
Response 1: Thank you again for offering us the possibility to improve this paper.
Point 2: I would like to ask you to take a critical look at the following aspect in your discussion: Posture is the interaction of bones, tendons and muscles. When standing and sitting, muscular activity changes the curvature of the back and thus also the measurable kyphosis and lordosis angles. You should explain in more detail how you were able to ensure that the children did not change their trunk posture due to muscular activation during the measurements. I think that this should be clearly addressed in the limitation section, because it is a limiting factor that we generally have in posture measurements (especially in children) and that can limit the accuracy of a classification of the measurement results.
Response 2: Thank you for the suggestion. The idea has been included and it has been explained at the end of the discussion section as part of the study strengths and limitations.
Furthermore, some sentences in the Materials and Methods section have been added to explain with more detail the protocol and how to maintain the quality and to perform accurate measurements.
Point 3: line 91: "...injury. While..." --> "...injury, while..."
Response 3: This sentence has been modified.
Point 4: line 103: "It was a descriptive study." --> "The presented study was a descriptive study."
Response 4: This sentence has been modified.
Point 5: line 206: "...Santonja [14]after..." --> "...Santonja [14] after..."
Response 5: This sentence has been modified.
Point 6: Table 2 & 3: what is the difference between the 'static' and 'dynamic' subclasses? I cannot find an explanation in the text.
Response 6: The explanation in the text has been added.
All subclasses are in relation to the position or positions where misalignment occurs. Specifically, the "static" subclasses are in relation to the slump sitting position or to the standing position or in both positions; the "dynamic" subclasses are related to the forward bending position, while the "total" subclasses are referred to when the misalignment occurs in the slump sitting position and in the forward bending position.
Point 7: line 286: "...these studies only assessment..." --> "...these studies only assessed..."
Response 7: This sentence has been modified.
Point 8: line 362: "...70.45% and 89.06% of schoolchildren..." - boys and girls?
Response 8: Yes, it refers to both sexes. This sentence has been modified.
We hope that all the changes can address your considerations.
Kind regards,
The authors.